# Self-Esteem, Meaningful Experiences and the Rocky Road—Contexts of Physical Activity That Impact Mental Health in Adolescents

**DOI:** 10.3390/ijerph192315846

**Published:** 2022-11-28

**Authors:** John Murphy, Bronagh McGrane, Rhiannon Lee White, Mary Rose Sweeney

**Affiliations:** 1School of Arts, Education and Movement, Dublin City University Institute of Education, D09 Y590 Dublin, Ireland; 2Student Services Department, Dundalk Institute of Technology, A91 K584 Dundalk, Ireland; 3School of Health Science, Western Sydney University, Penrith, NSW 2751, Australia; 4School of Nursing, Psychotherapy and Community Health, Dublin City University, D09 NRT0 Dublin, Ireland

**Keywords:** wellbeing, exercise, mechanisms, resilience, sport, qualitative, opportunity

## Abstract

Abundant evidence shows that physical activity benefits adolescents’ mental health and wellbeing. Quantitative evidence has shown that adolescents engaging in leisure time physical activity, a number of sports, and team sport, display better mental health outcomes than their peers. The specific contextual factors that contribute to increased mental health and wellbeing through physical activity are, as yet, unconfirmed. The purpose of this study was to identify the contexts of physical activity and sport that positively impact mental health and wellbeing as perceived by adolescents. A sample of 58 adolescents participated in 13 focus groups discussing various factors related to physical activity, sport and mental health. Participants brought an object that represented physical activity and an image that represented wellbeing to each focus group to aid in the discussion and representation of both. An inductive thematic analysis was conducted on transcripts of the focus groups using a six-phase approach. Five key themes were identified: (1) achievements and improvements leading to increased self-esteem; (2) the importance of meaningful experiences, a sense of belonging and contributions to identity; (3) development of resilience and responding to setbacks; (4) social connectedness and relatedness, and (5) an opportunity to experience mindfulness, distraction and flow-states. In order to enhance and support wellbeing through physical activity, adolescents should be encouraged and provided with opportunities to engage in enjoyable activities with people with whom they experience a sense of belonging, where there is an opportunity to experience mastery and improvement and that includes an element of autonomy or choice.

## 1. Introduction

Mental health disorders are the second leading cause of the global burden of illness, and prevalence is growing rapidly [1]. Lack of physical activity is well recognised as a key risk factor for the management and prevention of mental ill-health, including anxiety and depression [2], and recent physical activity guidelines have acknowledged mental health benefits for those who achieve the recommended amount of activity. However, physical activity recommendations only detail the volume, intensity and type of activities that should be undertaken, while no reference is made to contextual factors that may lead to more optimal mental health benefits [3]. Due to the inability to identify an optimal dose, recent research has shifted to the context or life domain through which it is engaged. For example, leisure-time physical activity and active travel have been found to be positively associated with positive mental health, while leisure-time physical activity and sport were inversely associated with mental ill-health [3]. However, physical activity was not consistently associated with lower mental ill-health across domains, as work-related physical activity was positively associated with mental ill-health [3]. These results highlight the need to investigate the most appropriate contexts for improving mental health and wellbeing.

Current evidence suggests both physiological and psychological factors impact mental health through physical activity [4], with one hypothesis proposing that psychosocial mechanisms are largely at play, including self-esteem or self-efficacy, physical self-perceptions, social connectedness, and mood and emotions. Many theoretical frameworks have further proposed that optimal psychological functioning, or wellbeing, is achieved through satisfying basic psychological needs for social connectedness, autonomy, self-acceptance, environmental mastery, and purpose in life [5,6]. Indeed, physical activity provides an opportunity for social interaction (relatedness), mastery in the physical domain (self-efficacy and perceived competence), improvements in appearance (body image) and overall self-perceptions (body image), and independence (autonomy) [5]. Participation in physical activity may also lead to improved task self-efficacy, which initially generalises to a broader self-concept and then to global self-esteem [7]. However, physical activity may also have a negative impact on mental health outcomes in adults [3], adolescents and children depending on the specific context and circumstances [8,9].

Because increasing physical activity in and of itself is not likely to guarantee greater mental health and reduced mental ill-health, contextual factors are crucial [3]. Gaining knowledge of the specific factors that influence the relationship between physical activity and mental health, particularly within the leisure-time domain, will help to lead to the development of contextually tailored interventions and physical activity guidelines and improve the effectiveness of physical activity as a prevention and treatment method [3]. Therefore, the primary aim of this study was to examine how physical activity holds different associations with perceived wellbeing in different contexts. More specifically, we also aimed to identify which aspects of leisure-time physical activity best support wellbeing. Secondary objectives included identifying the aspects of leisure-time physical activity that best support wellbeing, identifying the aspects of team sport that may offer a “protective effect” against symptoms of mental ill-health, and ascertaining if experiences of leisure-time physical activity and their relationship with mental health, differ between males and females.

## 2. Methods

### 2.1. Participants

Participants (*n* = 58) were post-primary students from five schools in the southeast of Ireland and included both males (*n* = 27) and females (*n* = 31) aged 16–18. Each participant attended one of thirteen focus groups. Focus group interviews were subdivided into groupings of adolescents who played a team sport, those who played an individual sport, and those who identified as largely inactive outside of school. Two participating schools were all-male, two participating schools were all-female, and one was co-ed. Details of school, participants and groupings can be found in Table 1. Further groups were originally planned in another co-ed school, although the authors felt data saturation had been reached, having read notes and transcripts from the interviews that were conducted as part of this project. Participants engaged in a range of physical activities and sports, including hurling/camogie, Gaelic/ladies football, rugby, basketball, soccer, athletics, dance, gymnastics and rowing.

### 2.2. Design and Procedure

Following ethical approval by the Dublin City University Ethics Committee (DCUREC/2019/107), schools were purposively selected from a previous national questionnaire [10]. Invitations were sent to school principals. Once consent was provided by each Principal and school management, the lead researcher spoke with class groups of 4th or 5th-year students about the study and invited participants to take part. Students in their 4th and 5th Years are in the third and second last years of post-primary education in Ireland. A minimum of 60 min timetabled physical education lessons are compulsory for all students at this stage of school. Signed parental consent and participant assent were secured prior to any data collection taking place. The lead researcher conducted the focus groups on the grounds of each participating school. Participants were asked to bring to the focus groups an object that represented physical activity and an image that represented wellbeing for them. These aided in the discussion of their perceived connections between physical activity and wellbeing. Participants sat in semi-circles around three to four grouped tables, with the lead researcher at the centre of the semi-circle. During focus groups, participants were asked one by one about how physical activity may hold different associations with perceived wellbeing in different contexts, how physical activity may best support wellbeing, and how physical activity may negatively impact wellbeing. Participants were encouraged to provide personal anecdotes of their individual experiences. Focus groups were recorded on a digital voice recorder for subsequent transcription and analysis. Notes were kept by the lead researcher throughout each focus group for use in subsequent analysis. The lead researcher facilitated, transcribed, and analysed all focus group interviews.

### 2.3. Data Analysis

In order to help understand adolescent perceptions around the relationships between physical activity in different contexts and perceived wellbeing, inductive thematic analysis was conducted on focus group transcripts using Braun and Clarke’s [11] six-phase procedure. In line with guidelines, transcripts were examined, and initial codes were identified and refined by the authors as nine candidate themes were proposed. Establishing internal homogeneity-external heterogeneity in a manner proposed by Braun and Clarke [11] is important at this stage, so extracts that fitted into multiple themes were collapsed into a higher-order theme, and a new candidate theme was generated. As the stages progressed, some extracts were removed as stronger extracts that demonstrated each theme were used as examples. This pattern was repeated until there was a consensus between the authors that there were distinct themes. Five themes remained following this process, and each theme’s title was amended to reflect the study’s data and literature about contexts of physical activity that support wellbeing.

## 3. Findings and Discussion

This paper explores how physical activity holds different associations with perceived wellbeing in different contexts for adolescents, with a specific focus on the identification of which aspects of leisure-time physical activity best support wellbeing. Such results provide valuable insights for future physical activity recommendations. Five higher-order themes and fifteen sub-themes were used to convey the feelings of participants towards their experiences of physical activity and sport. These are summarised in Figure 1.

## 4. Achievements, Improvement and Self-Esteem

### 4.1. Achievements and Improvements Leading to Increased Self-Esteem and Confidence

The majority of participants noted the importance of, and enjoyment derived from, improvements in fitness, skill or all-around ability. Each participant was asked what the most enjoyable aspect of engaging in physical activity and sport was, with improvements over time being the most common response:

“*I think definitely the chance of getting better at things. I used to hate running but I got into it and I started to like the sense of achievement after finishing it and to notice that you could get faster or you could run a bit longer and you were starting to find it easier, as you were getting fitter and getting better at it. I just really liked that*.”(Inactive male)

Physical activity appears to provide opportunities to master skills, and this sense of mastery facilitates enjoyment on a regular basis which is essential to wellbeing. Follow-up questions focused on probing why improvements are enjoyable and how they may have further positive effects on wellbeing. Increases in self-esteem that carried over to other aspects of life were key:

“*Well, I do Muay Thai. I have done it for six years. [pause] I had struggled a lot with self-esteem when I was younger and started. Through it, it helped me gain confidence in my abilities and what I can do if I set my mind to something*.”(Individual activity male)

The confidence gained from physical activity could also help adolescents through other pursuits: “*I think that sense of achievement really boosts your confidence for the rest of the week*.” (Individual sport female).

### 4.2. Perceived Competence and Fear of Failure

Participants who self-identified as inactive highlighted some of the barriers that discouraged them from engaging more in physical activity. One of these barriers was a fear of not performing or not meeting the standards of others, which in turn led to less enjoyment:

“*Yeah, definitely, especially in school, even with PE, people are very demanding, and if you’re not meeting their expectations they’re kind of judge-y, and it just makes it less enjoyable*.”(Inactive male)

A lack of confidence in their own ability, or perceived competence, was also highlighted as a barrier to participating but also in deriving greater enjoyment from physical activity and sport:

“*I think you have to have some sort of interest; you have to like what you’re doing. I would not be able to play hurling for my life, so I think I would just be so caught up in my own head what everyone else is thinking. If I was to run for a ball, in my head I’d be like, ‘There’s no point you running for it because you won’t be able to get it off someone’ Or someone with more skill will come and take it*.”(Inactive male)

## 5. Meaningful Experiences, Identity and Purpose

### 5.1. Sense of Belonging or of Representation

Participants were asked why adolescents who engage in team sports have reported higher levels of wellbeing in the past when compared to inactive adolescents or those who participate in individual activities. The sense of belonging to a team or group was immediately highlighted by all: *“Being a part of a team in general because you have that sense of belonging and community as well, like being part of something.”* Being part of, or representing a wider group, was identified by all participants who engaged in sport as important, along with the sense of pride it brought when they represented their local community:

“*I’d always say my club jersey. Because it’s kind of like, yes, it’s a great achievement to go and play in county teams and it’s great representing your county, but when it comes to a championship and they give you the jersey, you get a shiver down your back when you’re putting it on. It’s like you’re playing for everyone now, it’s not just you. It’s bigger*.”(Team sport male)

The sense of belonging or representation was also highlighted by participants who engaged in sports largely categorised as an individual, such as athletics, dance and gymnastics:

“*I brought the singlet that I wear when I’m racing with the club. It represents it. When I’m wearing it, it’s nice to be able to represent the club… I’m proud of who I run for and try and represent them as much as I can*.”(Individual sport male)

It was important to note that a sense of belonging is not automatically felt or created but should be worked on and developed through specific activities.

### 5.2. Development of Personality, Purpose, and Sense of Self

Engagement in sport helped many of the participants guide the development of their personality traits and form a purpose in life:

“*Like, without sport, I was very lost as a person. I came up from Cork, about three years ago. Without sport, I would only have school to join in with friends. Sport is a massive thing to have, just to have something in common with the lads here… If I wasn’t actively playing sport, I don’t know where I would be in life*.”(Individual sport male)

Engagement in sport also provided structure to their week and helped them to remain focused on goals over the longer term:

“*I think they just go hand in hand together. Without sport, I wouldn’t be the same person I am, definitely, without sport. I wouldn’t be as happy as I am, or as much friends as I have, and all that. It gives me something to do, wake up every day, and go out, and know that I am doing something with myself. If I didn’t have sport, I would just be sitting around doing nothing. I’d be wasting…*”(Team sport male)

In two groups, participants also noted the importance of having other outlets in life as well as sport. This can be achieved through playing a number of sports or complementing physical activity with other non-sporting hobbies such as music:

“*Your sport can obviously be very important to you but you have a life outside of your sport. Don’t let your sport take over how you feel about yourself in the rest of life. Try to pick up another hobby that’s completely different and don’t lose a part of yourself to your sport*.”(Team sport female)

### 5.3. Intrinsic and Autonomous Motivation

The importance of framing physical activity experiences as an opportunity was also highlighted by the majority of participants in team sport:

“*There’s no point in playing a sport if you’re not happy doing it. I look forward to training. If you’re thinking, ‘I don’t want to go to this training,’ and all that, then you’re going to go down there and have a bad day, have a bad training. You have to be positive if you’re going down*.”(Team sport male)

Internal and often intangible rewards, such as feelings of accomplishment, were highlighted by both team and individual activity participants as important motivators in remaining with an activity and giving their full effort:

“*The reward at the end because it’s hard to stay motivated. After completing a hard session you feel good about yourself and that’s the biggest reward you can give yourself. Sometimes you’d be stressed or nervous about a hard session then when you complete it you feel even better again. You’ve kind of shown yourself that you’re capable*.”(Individual sport female)

Participants from all groupings also highlighted the difference in the enjoyment and perceived mental health benefits of engaging in intrinsically motivated physical activity as opposed to extrinsically motivated physical activity.

“*I feel like it improves your mental health, if it’s an enjoyable activity that you’re doing, it’s just kind of like you’re doing it at your own will and at your own pace. But I think it’s depending on if there’s someone telling you what to do, I think that kind of like makes it more or less enjoyable*.”(Inactive female)

### 5.4. Expectations and Pressure to Perform

Participants who predominantly engaged in individual sports, especially those with stereotypical body image expectations, noted the pressures imposed on them to look a certain way:

“*There’s a lot of pressure to be the tall skinny girl. Even though my ballet school is very accepting of everybody, but there’s still the social standard, I guess, lingering in the back of your mind, even if you don’t realise you’re thinking about it, it’s always there regardless of what you’re doing*.”(Individual sport female)

One individual noted that expectations to perform well in a sport could often place pressure on adolescents to overly focus on that aspect of their lives and not develop other areas due to the pressure to consistently perform:

“*Say, for instance, you have a fella, he’s an unreal hurler or Gaelic footballer. His entire life just circulates around the fact that he is a really important cog in this team. That ends up becoming… like it becomes their only personality trait, or their only go-to. If your fella is making teams all your life, and you don’t get the team then, you don’t know what to do with yourself*.”(Team sport male)

Societal expectations can often deter some adolescents from pursuing other interests and hobbies, as was also highlighted by another participant:

“*I would be a bit of a mixed bag when it comes to sports. I would be into music and arts and stuff. I feel like, definitely in the sports world, they don’t really go hand in hand with each other. Playing, singing and acting, I feel like, they could be considered more feminine things to do. There is a bit of stigma that fellas that are involved in that could be no good at sports, or no good at putting a tackle in. I feel like it is silly. Your other interests and passions have nothing to do with how good you are at something else*.”(Team sport male)

### 5.5. Negative Experiences and Lack of Purpose

A lack of meaning or purpose to certain activities was highlighted in almost every group: *“If we are playing for nothing, then we are training for nothing, as well.”* The impact of adults on adolescents’ teams was consistently highlighted as having a negative impact due to both coaching and administration. The culture or lack of camaraderie that was fostered within a team could lead to less enjoyment by the group as a whole:

“*It depends on the people you’re around. For example, I do my football and it’s really good for your wellbeing but then you have your coaches and they can really put you down… They made everyone so competitive where we were all against each other on the team when we were supposed to be playing together*.”(Team sport female)

One participant highlighted the negative influence of adults on community sport, especially in the selection of certain teams:

“*I definitely think the whole politics in schoolboy sports, from a young age, isn’t good for your wellbeing. It’s completely unnecessary. You have coaches, and they’re making serious decisions about whether this fella is an A player, a B player, when they are too young to make any evaluation. I feel like pegging someone in a category at a young age in sports is setting them up for an awful hard time in breaking out of their own shells in different aspects of life. When it comes to your own values, and you’re meant to be valuing yourself really highly, and you have adults who are meant to be the people that are guiding you, that are telling you you’re not good enough, or telling you that you can’t achieve what you want to achieve*.”(Team sport male)

Administration issues and lack of communication or consistency between teams for training or match times also caused frustration and lack of enjoyment and were commonly highlighted by female sport participants from all-female and co-ed groups:

“*You’re a teenager and like we play camogie and more likely play football as well and I just don’t understand why the club can’t work together and do one night camogie, one night football so it doesn’t clash. It doesn’t make any sense*.”(Team sport female)

## 6. Resilience, Setbacks and “The Rocky Road”

### 6.1. Transferable Psychological Skills

Task or activity-specific achievements, improvement and progressions were previously highlighted as having positive effects on self-efficacy and may transfer to global self-esteem. The psychological skills required and often developed through these achievements may translate to other non-physical activity-related aspects of life, as was highlighted by participants when asked about the potential benefits of engaging in more than one activity:

“*I think it’s important to learn and develop more mental skills across the different sports. Like individual sports would develop internal motivation and resilience while team sports would develop more communication and trust skills. If you’re doing a few different types of sports and developing a few different types of skills then you can transfer them to each other but also to normal life*.”(Team sport female)

Resilience and adaptability were highlighted by most team sport participants as being developed by sport and transferable to other aspects of life:

“*I think the fact that it’s not all easy. Like sport’s like life. There’s always ups and downs, but you always have to push through and be the bigger and better person and reflect on things, like right, okay, this is what I can change next time. I can do this better and this is what I did well and this is what… that’s not always straightforward. There’s always like things that change. So you have to adjust to it*.”(Team sport female)

The idea of commitment was outlined by the majority of team sport participants as important to the individual themselves in staying with a sport, but also in making a commitment to teammates:

“*But you’re also, especially team sport, when you know everyone else is going to be there, you’re like, well I can’t miss out. And not that you don’t want to miss out on anything but it’s just like you can’t let them down*.”(Team sport male)

### 6.2. Reacting to Losses and Setbacks

Learning to deal with or respond to losses and setbacks was highlighted by the majority of the team and individual sport participants in contributing to progressions over time and in developing resilience:

“*Yes, you have to learn the skills to deal with loss because you’re going to experience so much failure and setbacks in life. You have to learn to deal with it. And that’s a good thing in doing sports: it teaches you that you’re going to have setbacks. Then at least you’re prepped. If the first time you experience a setback is in school, you fail a test, you’re going to be completely doubting your abilities, whereas you learn that from a young age*.”(Inactive male)

Team sports naturally have an in-built support group through teammates. This makes it easier to actively seek social support, which in turn helps reflection on performance:

“*At first, you’re just annoyed at yourself, but then I think with the people around you, they kind of tell you, “Ah, yeah, you can learn from that and move on.” It will help you going forward. So it’s kind of both*.”(Team sport female)

Experiencing and reflecting on losses also helped adolescents to appreciate and derive greater value from successes when they do come:

“*And I think when you lose as well, when you do win it makes it feel even more rewarding because say if you just win all the time, that would eventually just start to lose its meaning, so I think when you lose once in a while it does really help to push you and do better and to help you make sure that you’re on the right path*.”(Team sport female)

Irish adolescents who participated in multiple activities were found to have higher levels of wellbeing than those who participated in one or none [10], while internationally participating in sport in three to five different settings was associated with better wellbeing compared to participation in a single setting or sport [12]. When asked about the potential benefits of participating in multiple sports, the majority of participants highlighted the opportunity to redeem themselves or atone for previous poor performances:

“*I think with multiple sports and you do something bad in one like losing or a bad performance then you have an opportunity to go into a different one and just try again or kind of make up for it. Like if you lose football on Sunday and then go to basketball Tuesday night like they don’t know how you did in the football so it’s just a clean slate straight away. It just helps you move on from the bad stuff and lets you build up your confidence again fairly quickly. A chance to redeem yourself for yourself*.”(Team sport female)

## 7. Social Connection and Relatedness

### 7.1. Shared Experiences

Second to improvements and progressions over time, “*winning with friends*” was highlighted as the most enjoyable aspect of engaging in a team sport. Sharing the experience of winning, losing and progressing over time was said to be more enjoyable than on your own:

“*The best thing is winning but celebrating it as a team together. We have that one big thing in common and can share the experience, especially after the few months of working together*.”(Team sport male)

Some suggested that enjoyment may last longer when it’s shared with others:

“*I suppose if something goes well with a team sport you’re all there together and you can all celebrate together, whereas by yourself I know you’ll have your family and friends or whatever but it’s only you who’s done that, whereas a team, it lasts a bit longer as well*.”(Inactive male)

Sharing the experience with friends and teammates can also serve as a motivator:

“*No, it’s important to win with your friends and your teammates. When you’re playing team sports, it’s not just about you. It’s for your friends and the team. You go the extra mile for the people on your team*.”(Team sport female)

The shared experience of simply moving forward and trying to get better also served as a source of enjoyment for some adolescents:

“*I think being with other girls and the chance to socialise but also to socialise with people who are pretty much on the same mission. We’re all there for the same reason, to get better, and that’s created a bit of a bond and we’ve created an environment where we’re just trying to do the best for the team with no judgement. That environment is just really enjoyable to be a part of. All of us just trying to be a little bit better together*.”(Team sport female)

The enjoyment of and perceived benefits from shared experiences were not exclusive to team sport participants. Dancers and gymnasts also spoke about how they can experience similar enjoyment when their individual effort combines with others to execute a group performance successfully:

“*It’s yourself because you practice a lot, and then when we all come in together and we all have the dance right, I think it’s kind of personal but it’s also a group effort as well, because when we’re in class and they’re telling us to go home and practice, that’s the personal part of it, whereas you practice and you try and you keep looking at yourself and being like “Is this the right way, is this not?” and then you come back the next week and then everybody has practiced and we all look good together, so it’s an achievement for yourself but achievement for everybody else too*.”(Individual sport female)

### 7.2. Facilitator of Connection

Engagement in sport and physical activity can facilitate a connection between other adolescents with similar interests: *“It’s like a portal you go through to meet your friends”.* Sport serving as a facilitator of connection was evident in both females:
“*I would be very close to my friend group. Sport also provides support as it’s a chance to meet some other girls and provides a distraction. I love the basketball girls as we’re all there for the same reason and that helps to bond us together*.”(Team sport female)
and males, who also stated it was important in helping them develop a connection with teammates who were a bit older or younger:

“*I made most of my friends through Gaelic football and through school. The school team has also helped to pull us together with TYs and 6th years… We just had fun or had to communicate at training. Then when we’d meet on the corridor we’d say hello and just start meeting up more often outside of school and football*.”(Team sport male)

Further expansion on why physical activity with others is important. One participant stated: *“It’s like people respond to people, and it’s better than doing it alone and it’s something you can share what you’re into with other people.”* (Team sport male). Another stated the importance of making friends through sport, and the impact that has had on other aspects of life:

“*I think that without sport, we would all be completely different people. We wouldn’t have the same group of friends. We wouldn’t all have the same passion for things as everyone else. We wouldn’t be… I don’t know what the word would be, we wouldn’t connect with people as much, because you’re not into the same things as each other… If you’re not connected with people, you have to have something connecting, the way you have sports to play with people, and you know what they’re doing, and we’re doing. But if you’re not all in the same thing, then there’s nothing connecting you with them*.”(Team sport male)

Discussions around sports acting as a facilitator of connection led to some notable points on how connections can also be formed between people and personalities who may not necessarily have much in common:

“*Say you’re on a team with some people that you like, and some people that you don’t like, when you win, there is definitely a window for a couple of hours where you are all just delighted to be with each other. It’s like a little window comes down, where you’re all the same, for the day. Then you can go back to not liking each other tomorrow*.”(Team sport male)

One female participant highlighted the importance of a sport with other people as opposed to individual activities:

“*I think like, not a social aspect or having a craic, I feel like, if friends weren’t there, it just wouldn’t…because, I’m just doing it out of pure enjoyment and enjoying spending time with my friends, doing the sport that I love and having something to look forward to. If I didn’t play sport, you’d be a bit lost. Like, even if you do play an instrument, or run, I just think it would lack enjoyment if you didn’t have this team sport to play with your friends and enjoy*.”(Team sport female)

## 8. Mindfulness, Distraction and Flow

### 8.1. Peace, Calm and Mindfulness

Participants who did not regularly engage in physical activity highlighted some potential benefits from the few positive experiences they had:

“*For me it’s just listening to music and looking at everything, just cycling away, just taking in everything I see. Because I’m not thinking about anything, I’m just listening to something. I’m not thinking about, ‘I have an assignment due,’ or I’m not thinking what time I’m going to bed at; stuff that I would normally overthink about*.”(Individual sport male)

A number of activities were suggested as helping to experience a sense of peace or calm, such as running, cycling, and swimming by participants who were both active and inactive:

“*When you’re swimming it’s a breakaway, you’re in your own headspace, there’s no-one there telling you what to do, it’s just like you’re there to swim. And when you’re swimming, to me it’s gotten so natural now that I don’t think about swimming, I think about what’s going on in my head?*”(Individual sport female)

Another participant suggested listening to music while being active added to the experience: *“I find listening to music to be a really immersive experience then if you’re overwhelmed it just takes your mind away from it all and you kind of forget about it.”* (Team sport female). Getting a break from school work or other life stresses was identified as a real positive of being active by the majority of team sport participants:

“*Because like it’s not just about playing hurling, you get away from everything—you don’t really think about anything like from school and if you are having trouble at home, you just go away, just hurl and you don’t need to think about anything else*.”(Team sport male)

Other participants from individual activities noted how they found it easier to deal with a problem or to concentrate on schoolwork after physical activity:

“*Well, sometimes, say when I’m running, at the end of it my head is cleared and I can look at it in a different way because usually before I’m running I’ll be thinking a lot about it. But then when I run it just helps to free that up*.”(Individual sport female)

### 8.2. Outdoors and Green Spaces

When asked, each group suggested there were more potential benefits from outdoor activity compared to indoor activity:

“*Physically you would, but emotionally and mentally I wouldn’t say you would because for me it’s being outside in the air and being surrounded by other things more than just being in a room sitting on a bike that goes nowhere doing nothing*.”(Inactive male)

Greater enjoyment was also derived from being outdoors:

“*It’s all outside and training, like, I don’t like being inside and sitting down or watching telly or playing PlayStation. I like being out and being active and doing sports, that’s just when you are doing that, your wellbeing is better—well, mine is, anyway*.”(Team sport male)

The connection with nature was highlighted as an important contributor to greater enjoyment levels: *“Yes, I think it boosts your serotonin definitely because all the air and the animals around you make you feel a bit more connected to nature.”* (Individual sport female) along with noting how large and vast the world around us is:

“*So we’ll go hiking in the mountains, you’re out in nature and I know it’s a bit of a cliché but it does help you feel at peace because you’re on top of this huge mountain looking around and seeing how the world is so big and you’re so small and the problems seem less big then, because you’re walking*.”(Inactive male)

### 8.3. Experiencing Flow States

Many participants described experiences similar to flow when speaking about engaging in their favourite physical activity: *“I think it feels like, everything else takes a back seat. You’re living right there, right then. Nothing else matters. That is exactly what you’re doing. That’s all you can think about.”* (Individual sport female). ‘Flow-like’ states allow performers to block out all distractions and focus completely on the task they’re currently engaged in:

“*A lot of it is about blocking out the constant noise and bright lights of the world. That is exactly what happens when you do a sport, regardless of what the sport is. You get in the zone, and everything else fades away, and you get some time to just focus on what you’re doing*.”(Individual sport male)

When asked about what exactly they’re thinking about or focusing on, all participants struggled to put it into words:

“*You’re just not thinking of anything. You’re thinking about the match. Nothing else matters. Nothing else that has happened that day or that week will matter. That doesn’t even come into it. You’re just thinking about what you’re doing… There’s no command of it. It’s just, what happens, it happens in the moment*.”(Team sport male)

Both team and individual sports performers spoke about the enjoyment of ‘flow-like’ states:

“*I feel like when I dance it’s just a breakaway from reality. It’s like I’m in my own mind-set, and when I look at this picture it feels the same; she just looks as though she’s in her world, she’s having the time of her life, and to me my dancing’s that, that’s what I feel*.”(Individual sport female)

## 9. Male and Female Differences

Throughout the analysis of transcripts and field notes, some notable differences in responses between male and female participants became apparent. Females discussed physical activity as an opportunity to de-stress from life’s problems, mainly school or academic-related, so they could return to the situation or task with a fresher perspective, while males spoke of simply wanting to avoid or distract from the situation. Most interestingly, females elaborated on the same sense of belonging and being with teammates as an important time to be with friends and develop relationships, while males simply spoke of ‘not being alone’. Differences in motivational factors between males and females for engaging in physical activity with others have received little attention in the past and warrant further exploration. If females require a greater emphasis on the sense of belonging aspect of sports and an opportunity to incorporate other pursuits, such as academics, into their athletic career, then this should be facilitated where possible.

## 10. Discussion

This study sought to understand adolescent perceptions around the relationships between physical activity in different contexts and perceived wellbeing. Insights were gathered through semi-structured focus group interviews from a variety of adolescent groups, including those active in team sports, active in individual sports/activities and adolescents who self-identified as inactive. Focus group interviews were conducted in all-female, all-male and co-ed settings. The major themes identified include the importance of achievements and improvements over time, the importance of meaningful experiences in physical activity and sport, the development of resilience through losses and setbacks, the opportunity to develop social connections with others, and the opportunity to experience mindfulness or flow-like states. This is the first study of its kind to qualitatively build on previous quantitative findings [10,13] around adolescents’ perceived relationships between contexts of physical activity and wellbeing.

Increased self-efficacy has been identified as a possible mechanism for increased global wellbeing in adolescents [13] which suggests increasing task-specific self-confidence through physical activity or sport may have positive benefits for other aspects of life for adolescents. Task-specific self-confidence, or self-efficacy [14], influences persistence, thoughts, stimulation and behaviour as self-perceptions lead to positive experiences and may be transferable across life domains. While physical appearance is generally a greater predictor of global self-worth in adolescents than athletic competence [15], Harter [16] argued that the contribution of domain-specific self-worth to global self-worth, and therefore wellbeing, is a function of the importance an individual places on each domain. Therefore, increasing an adolescent’s perceived value of physical competency could enhance the ability of PA to increase global self-worth. Higher levels of perceived competence have previously been shown to facilitate adolescents in adapting to stressors while being active [17] as they are more confident in their own physical abilities and, therefore, better prepared to deal with setbacks in some physical activity contexts. Self-presentation [18] has also been suggested as a negative influence on mental health through a lack of perceived competence, although self-presentational concerns are often appearance-related. Concerns that one will be unable to make the desired impression due to a lack of skills or strength required to perform the task at hand can also arise [19].

The need to form and maintain interpersonal relationships is at the core of satisfying the need for belonging. A sense of belonging among teams or groups satisfies a basic psychological need as adolescents who experience a greater sense of belonging have stronger inner resources, a sense of identity, and intrinsic motivation [20]. Current evidence is conflicted in regard to physical activity with others, such as performing exercise ‘beside’ someone as opposed to ‘with’ them. Identifying as part of a group, often working toward a common goal and sharing the experience is identified here as particularly important in contrast to simply performing exercise with others in the same location. A sense of belonging is often characterised by feelings of inclusion, connectedness, and support [21]. A previous qualitative study examining the relationship between physical activity and wellbeing noted how participants spoke of physical activity as an “opportunity” [22]. Participants who engaged in team or individual sports spoke of going to training or competitions in a similar way. Intrinsically motivated individuals are more likely to remain engaged in an activity for the longer term and derive greater enjoyment from the same activity [5]. Controlled motivation is a likely contributor to lower derived enjoyment and reduced mental health benefits as controlled reasons for participation include feeling pressured to do a sport or activity due to cultural background, family history, or being forced to participate, either by parents, coaches or teachers. These findings are in line with self-determination theory which suggests autonomously motivated behaviours are more likely to be associated with greater psychological wellbeing compared to activities which are carried out due to controlled motivation [5,23] and confirm previous findings from a physical education setting [24].

The expectations of looking a certain way in a sport that also celebrates the individual over the team or group can also make it difficult to cope with some of the body image pressures. Self-presentational concerns [18] have already been highlighted as a contributor to lower participation in and enjoyment of physical activity. Self-presentation is often linked with social physique anxiety [25] which is not associated with any frequency or duration of exercise but through interaction with situational factors [26] or contexts related to the display of physique. This interaction may be manifested through the selection of physical activity settings based upon factors such as the social context and normative exercise attire [19], which may emphasise the shape and figure of some participants. Steele and Aronson [27] propose a phenomenon known as ‘stereotype threat’ that refers to the perceived risk of confirming, through one’s behaviour or outcomes, negative stereotypes that are held about one’s social identity. They suggest stereotypical external views can deter individuals from pursuing other interests, in this case, music and arts. Dee [28] proposes an alternative model of social identity [29] that suggests individuals experiencing stereotype threat do not necessarily feel they personally do (or should) subscribe to the stereotyped traits of a particular social identity, but it is the apprehension that others view them through the lens of a certain stereotype that impacts on their behaviours.

A lack of fulfilment or meaning has previously been cited as a reason for dropout in adolescent sport [30] and may ultimately lead to dropout stemming from lower enjoyment or engagement on a consistent basis. This is consistent with previous investigations of barriers to physical activity in adolescents [30,31,32], although the current findings suggest it may also be a barrier to the enjoyment or development of wellbeing through physical activity. Inequalities in terms of treatment or stereotypes have previously been highlighted in research focusing on female coaches, as they were found to be at greater risk of reduced wellbeing due to unfair treatment [33,34]. Although this specific issue has not been previously addressed in adolescent athletes, it does warrant further attention in terms of the allocation of equal access to resources and facilities regardless of gender.

MacNamara et al. [35] outlined a number of ‘above the neck’ skills known as psychological characteristics of developing excellence (PCDEs) shown to play a crucial role in the realisation of potential. PCDEs allow young performers to optimise development opportunities, adapt to setbacks and effectively negotiate key transitions along pathways to excellence and throughout life. These PCDEs are often developed through the various losses and setbacks that are inevitable when playing sports. The various ‘setbacks’ or challenges that naturally occur through involvement in sport are deemed essential to optimal performance later in life and referred to as “The Rocky Road” in talent development [36]. This “Rocky Road” is crucial in developing many PCDEs in adolescents, most notably resilience. The concept of resilience refers to findings that some individuals have relatively good psychological outcomes, despite exposure to acute or chronic stressors that are associated with negative outcomes [37]. Definitions of resilience vary, although most incorporate three pivotal concepts: stressors, positive adaptation and protective factors [38]. Losses and setbacks are obvious stressors, while the support of teammates, an ability to take on board feedback, and realistic performance evaluation, another PCDE, all serve as protective factors leading to positive adaptation. Learning how to reflect on losses and setbacks through the lens of a performance evaluation helps to turn each experience into part of a learning process instead of one individual outcome. Evaluating current performances compared to previous ones is also important in developing a growth mindset [39]. Irish adolescents who participated in multiple activities were found to have higher levels of wellbeing than those who participated in one or none [10], while internationally, participation in three to five activities was associated with higher wellbeing [12]. The results suggest this may be due to having another opportunity to redeem themselves after a poor performance or result.

Studies in adults have found that when physical activity is completed with others, there is a lower likelihood of developing depressive symptoms [40]. Satisfying the need for belonging depends on frequent personal contact and interactions with other individuals. Adolescents who have strong and well-established social relationships are less inclined to seek out additional bonds than adolescents who are socially deprived [41]. Sport provides an opportunity to develop social relationships through frequent contact, supportive environments and expectations around a commitment to each other. Participating in physical activity with or alongside others is unlikely to provide the same benefits as when adolescents feel connected to each other. Satisfying the basic psychological need for relatedness appears to influence whether physical activity experiences are associated with positive affect [9]. Therefore, it is an important contributor to the development of wellbeing through physical activity.

Adolescent-based studies have previously highlighted the importance of physical activity as a means to distract from everyday stressors and experience mindfulness [9]. Engaging in physical activity outdoors has demonstrated more favourable benefits to physical and mental health when compared to exercising indoors [42], which may be due to exposure to nature and the extra mindfulness benefits associated with exercising outdoors. Academic failures or pressures have been identified as risk factors against the development of wellbeing in Irish adolescents [43], and physical activity has been highlighted as an opportunity to distract from these pressures. Mindfulness may also be experienced when adolescents engage in ‘flow-like’ states when exercising. Csikszentmihalyi [44] found that adolescents experienced immense pleasure and serenity when engaged in flow states and reported higher levels of happiness with greater volumes of exposure, thus leading to increased wellbeing due to more time spent in flow states. ‘Flow-like’ states are experienced when an individual is deeply engaged in an activity. Nakamura and Csikszentmihalyi [45] suggest that motivation to perform or complete a task is highest when the difficulty of said task is matched by the individual’s abilities and skills. A ‘flow’ state or supreme enjoyment and engagement in the task is experienced when optimal challenge meets with suitable skill and ability.

## 11. Conclusions

This study is the first of its kind to qualitatively explore adolescents’ perceptions of how physical activity holds different associations with perceived wellbeing across different contexts. The participants in this study suggested that increases in self-esteem from physical activity may transfer across to other aspects of life, thereby improving global self-esteem and perceived wellbeing. The inevitable losses and setbacks in sport were highlighted as being important for the development of resilience in adolescents, while the development and maintenance of a sense of belonging were deemed important for enhanced enjoyment and continuity of a sport for many adolescents. Physical activity also provides adolescents with an opportunity to experience mindfulness and engage in ‘flow-like’ states which also contribute to enhanced wellbeing. In terms of negatively impacting wellbeing, physical activity experiences should reduce the focus on outcome-oriented goals such as winning and losing, comparisons to peers, focusing on body image or appearance, and forced involvement in physical activity or sport. Promoting autonomously motivated physical activity which satisfies adolescents’ psychological needs is likely to be the most effective method of enhancing and supporting wellbeing through physical activity.

## Figures and Tables

**Figure 1 ijerph-19-15846-f001:**
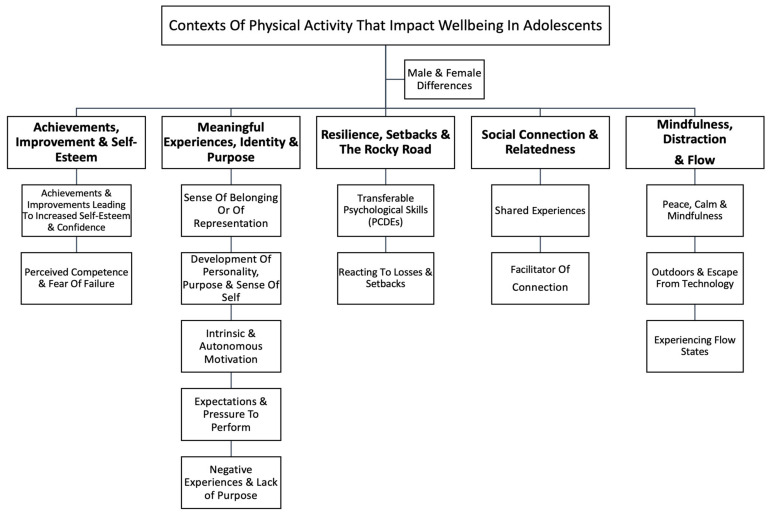
Adolescents perceived influences of physical activity and sport on mental health.

**Table 1 ijerph-19-15846-t001:** Groupings of focus group participants.

Group	All-Male		All-Female		Co-Educational	
	Groups	*n*	Groups	*n*	Groups	*n*
Team Sport	2	4, 5	2	5, 5	1	4 (2 male, 2 female)
Individual Sport/Activity	2	5, 4	2	4, 5	1	4 (1 male, 3 female)
Inactive (Self-identified)	1	4	1	4	1	5 (2 male, 3 female)

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
