# Peer review of "Self-Esteem, Meaningful Experiences and the Rocky Road—Contexts of Physical Activity That Impact Mental Health in Adolescents"

_ijerph, 2022, doi:10.3390/ijerph192315846_

Round 1

Reviewer 1 Report (Previous Reviewer 2)

I compared my edits and the changes made in the manuscript.

I am satisfied with the changes and approve this for publication.

Reviewer 2 Report (Previous Reviewer 3)

I think that my concerns have been addressed. 

This manuscript is a resubmission of an earlier submission. The following is a list of the peer review reports and author responses from that submission.

Round 1

Reviewer 1 Report

Thank you for the opportunity to review the manuscript that you submitted into the IJERPH for consideration. Overall, I found it to be an interesting read worthy of publication. However, from my viewpoint, there are some relatively minor edits required - please refer to the attached file for specific details. 

Author Response

Thank you for reviewing this paper. We are largely in agreement with the majority of comments and hope we have addressed them adequately in the updated manuscript. Your additions have added tot he quality (and hopefully value) of this paper. 

Reviewer 2 Report

Very well written manuscript, really impressed, good Job!

Please add how focus group was conducted and how many participants were on each focus group. 13 focus group for 56 total participants not justified. Typical focus group will consists of 6-8 people.

For the conclusion section: add how your finding impact mental health of adolescents...

Author Response

Thank you for reviewing our paper. All of your comments were valid and fair. We feel we have addressed everything and hope it is to your satisfaction.

We added a table that provides information on the number and groupings of focus groups.

We have added extra information to the procedure of the focus groups.

Also, while not in the paper, a previous study conducted by the lead researcher it was noted that adolescents were less likely to be open and honest when in groups larger than 5 while some became restless when more responses had to be waited for (again in groups of larger than 5).

The conclusion has been re-written.

Reviewer 3 Report

The relationship between physical activity and well-being, especially the psychological ones, is central to understanding what makes individuals do sports and, perhaps most importantly, continue doing sports. In this study, they conducted focus group interviews with both active in a team and individual sports as well as inactive, although there is only evidence for male inactive. It can show important results.

The result merged with the discussion becomes too long and thus unfocused. There are long quotes and long paragraphs interspersed with discussion and past references. As a reader, I eventually lose interest because it becomes too long-winded. As a reviewer, I understand that this feeling of wanting to stop reading is due to the article losing focus and, therefore, needs to be clarified here. Results would benefit from being separated from the discussion. That is, a review of how many quotes you need, and also to see a clear difference in the result regarding not only females and males but also the type of sport they practice. The inactive men their quotes are different from the others. However, what needs to be clarified is the representativeness of these quotes. Is it that some only had quotes from, for example, team sports, and that is why all the quotes are from this group or was this a coincidence? This may be relevant to show representativeness.

Regarding the method, I need a more detailed description of the focus groups. How many men team sports, individual sports, inactive? How many women are in team sports, individual sports, and inactive? To also understand the representativeness of the text. The focus groups are a mix of these individuals, or where they divided into team sports and inactive, and individual sports? How many schools were asked before you were selected? How many students were asked before you got this sample? Could there be a bias in this? This also comes to my next aspect. I miss limitations in the discussion. In conclusion, field notes reveal something that was not discussed in the method.

There is no overlap between purpose, which feels more quantitative, with the procedure, what you tell the participants and what you then use in the result. See the clips below. It could, of course, be about semantics, but it could also explain the feeling of confusion I experienced in the results section. What do the different lenses respond to?

From aim

Therefore, the primary aim of this study was to explore contexts of physical activity that have the strongest associations with perceived well-being and mental health in Irish adolescents. Secondary objectives included identifying the aspects of leisure-time physical activity that best support wellbeing, identifying the aspects of team sport that may offer a “protective effect” against symptoms of mental ill-health, and ascertaining if experiences of leisure-time physical activity and their relationship with mental health, differ between males and females.

From Procedure:

“participants were asked about connections between physical activity and wellbeing; how physical activity may improve wellbeing; how physical activity may negatively impact wellbeing”

From Result

To help understand adolescents’ thoughts and experiences of the relationship between physical activity and wellbeing

Author Response

Thank you for the very fair and worthwhile review. We agree with the majority of your points and hope we have adequately addressed them.

The merging of results and discussion was a contentious issue throughout the entire process but your review has swung it in favour of the separate camp. We also hope the addition of the table outlining the focus groups will explain their make-up more clearly.
